# Detection of Bovine Leukemia Virus in Argentine, Bolivian, Paraguayan and Cuban Native Cattle Using a Quantitative Real-Time PCR Assay-BLV-CoCoMo-qPCR-2

**DOI:** 10.3390/pathogens14101005

**Published:** 2025-10-04

**Authors:** Guillermo Giovambattista, Aronggaowa Bao, Olivia Marcuzzi, Ariel Loza Vega, Juan Antonio Pereira Rico, Maria Florencia Ortega Masague, Liz Aurora Castro Rojas, Ruben Dario Martinez, Odalys Uffo Reinosa, Yoko Aida

**Affiliations:** 1Instituto de Genética Veterinaria “Ingeniero Fernando N. Dulout” (UNLP-CONICET LA PLATA), Facultad de Ciencias Veterinarias, UNLP, Av. 60 y 118 S/N, La Plata 1900, Argentina; olimarcuzzi@gmail.com; 2Laboratory of Global Infectious Diseases Control Science, Graduate School of Agricultural and Life Sciences, The University of Tokyo, 1-1-1 Yayoi, Bunkyo-ku, Tokyo 113-8657, Japan; bao-aronggaowa607@g.ecc.u-tokyo.ac.jp; 3Facultad de Ciencias Veterinarias, Universidad Autónoma Gabriel René Moreno, Santa Cruz de la Sierra 00591, Bolivia; arlove@gmail.com (A.L.V.); juanantonio.pereira@gmail.com (J.A.P.R.); 4Agencia de Extensión Rural (AER Lules-EEA Famaillá), Instituto Nacional de Tecnología Agropecuaria (INTA), Ruta Provincial 301 km 32, Tucumán 4132, Argentina; florenciaortega40@gmail.com; 5Facultad de Ciencias Veterinarias, Universidad Nacional de Asunción, Acceso Avda. Mcal. Lopez a Campus UNA, San Lorenzo 111421, Paraguay; lcastro@vet.una.py; 6Facultad de Ciencias Agrarias, Universidad Nacional de Lomas de Zamora, Av. Juan XXIII y Ruta Provincial Nº 4, Lomas de Zamora 1832, Argentina; martinezruda@yahoo.com.ar; 7Centro Nacional de Sanidad Agropecuaria (CENSA), Carretera de Jamaica y Autopista Nacional, San José de las Lajas, La Habana 32700, Cuba; ouffor@gmail.com

**Keywords:** bovine leukemia virus (BLV), deltaretrovirus, endogenous retrovirus (ERV), molecular epidemiology, BLV epidemiology

## Abstract

Bovine leukemia virus (BLV), an oncogenic retrovirus of the genus Deltaretrovirus, causes enzootic bovine leukosis (EBL), the most prevalent neoplastic disease in cattle and a major source of economic loss. While BLV prevalence has been studied in commercial breeds, data on native Latin American cattle remain limited. This study assessed BLV infection and proviral load in 244 animals from six native breeds: Argentine Creole (CrAr), Patagonian Argentine Creole (CrArPat), Pampa Chaqueño Creole (CrPaCh), Bolivian Creole from Cochabamba (CrCoch), Saavedreño Creole (CrSaa), and Siboney (Sib), sampled across Argentina, Bolivia, Paraguay, and Cuba. BLV-CoCoMo-qPCR-2 assay detected BLV provirus in 76 animals (31.1%), with a mean load of 9923 copies per 10^5^ cells (range: 1–79,740). Infection rates varied significantly by breed (9.8% in CrAr to 83.8% in CrPaCh) and country (15.6% in Argentina to 83.8% in Paraguay) (*p* = 9.999 × 10^−5^). Among positives, 57.9% exhibited low proviral load (≤1000 copies), and 13.2% showed moderate levels (1001–9999), suggesting potential resistance to EBL progression. This is the first comprehensive report of BLV proviral load in Creole cattle across Latin America, offering novel epidemiological insights and highlighting the importance of native breeds in BLV surveillance.

## 1. Introduction

Bovine leukemia virus (BLV), a tumorigenic retrovirus belonging to the genus Deltaretrovirus, is the etiological agent of enzootic bovine leukosis (EBL), one of the most prevalent neoplastic diseases affecting cattle. Upon infection, BLV integrates into the host genome and may remain clinically asymptomatic in a condition referred to as the aleukemic state. Alternatively, it can progress to persistent lymphocytosis, a hematological disorder characterized by an elevated population of B lymphocytes, which may ultimately develop into B-cell lymphomas across various lymphoid tissues in infected animals [1,2]. Moreover, BLV-induced malignant cells are capable of infiltrating multiple organ systems, including the abomasum, heart, intestine, kidney, lungs, liver, and uterus. Clinical manifestations of BLV-associated tumors predominantly include gastrointestinal dysfunction, loss of appetite, weight loss, generalized weakness, lymphadenopathy, decreased milk yield, and reproductive inefficiency, each contributing to considerable economic losses within the livestock industry [3,4,5,6,7,8].

EBL was first documented in 1871 in Germany [9]. The virus initially emerged in the Memmel region of East Prussia, now Klaipėda, Lithuania, and subsequently disseminated across all continents, primarily via the international trade of breeding livestock. The prevalence of BLV infection exhibits considerable variability both among and within countries. Due to the implementation of successful eradication programs [10,11], most Western European nations, along with Australia and New Zealand, are either entirely free of BLV or maintain restricted zones of infection. In contrast, elevated BLV prevalence rates have been reported in both dairy and beef cattle populations across many countries outside these regions [12,13,14,15]. In Latin America, EBL was first recognized in Brazil in 1943 [16]. Estimates of BLV prevalence in Zebu, Taurine pure breeds, and crossbred cattle across Latin American countries range from below 20% to over 80% at the national survey, and from 0% to nearly 100% across specific breeds [17,18,19,20,21,22,23,24,25,26].

In Latin America, the dairy and beef cattle industries are well developed and contribute substantially to regional economies. The continent’s diverse agroecological zones support the breeding of various cattle types for dairy, beef, or dual-purpose production systems.

These include Zebuine breeds (e.g., Nellore, Brahman, Gir, Guzerat), Taurine breeds (e.g., Angus, Hereford, Holstein, Jersey), and composite breeds (e.g., Brangus, Braford, Girolando). Within the Taurine group, Creole breeds hold particular historical and genetic significance. These cattle are direct descendants of animals introduced by Spanish and Portuguese colonists during the late 15th and early 16th centuries. Following over 500 years of natural selection, Creole cattle have evolved into resilient populations exhibiting adaptation to local environmental conditions, extensive phenotypic variation (notably in coat coloration), elevated longevity and fertility, and notable resistance to endemic subtropical diseases. This includes reduced susceptibility to the tick *Boophilus microplus*, a prevalent vector of multiple bovine pathogens [27,28].

A range of diagnostic techniques has been developed and implemented for BLV, including serological assays targeting anti-BLV antibodies (such as agar gel immunodiffusion [AGID], enzyme-linked immunosorbent assay [ELISA], phytohemagglutinin [PHA], and radioimmunoassay [RIA]), as well as nucleic acid-based polymerase chain reaction (PCR) assays for the detection of the proviral genome (e.g., standard PCR, nested-PCR, quantitative real time PCR [qPCR], and direct blood-based PCR) [13,14,29,30,31,32,33,34]. Notably, the BLV provirus remains integrated into host cellular genomes even in the absence of detectable antibodies. Moreover, proviral load (PVL), which is the number of copies of a provirus, has been associated with BLV-associated disease progression [35,36,37] and transmission potential [35,38,39,40,41]. Accordingly, quantitative real-time PCR is recommended for accurate epidemiological assessments.

To date, ten BLV genotypes have been identified, initially through restriction fragment length polymorphism (RFLP)-PCR analysis and subsequently via partial or complete genomic sequencing [2,23,42,43], seven of which have been detected in Latin America [2]. The present study aimed to evaluate the prevalence of BLV in Creole cattle from Argentina, Bolivia, and Paraguay, as well as a composite local breed from Cuba. To our knowledge, BLV PVL has previously been assessed using the BLV-CoCoMo-qPCR-2 assay only in Bolivian Yacumeño Creole cattle, the Hartón del Valle Creole, and the Lucerna composite breed [23,44,45].

## 2. Materials and Methods

### 2.1. Sample Collection, DNA Extraction, and Plasma Isolation

Blood samples were collected from 244 adult cattle representing six distinct local breeds/populations: Argentine Creole (CrAr), Patagonian Argentine Creole (CrArPat), Pampa Chaqueño Creole from Paraguay (CrPaCh), Bolivian Creole from the Cochabamba Department (CrCoch), Saavedreño Creole (CrSaa), and the composite breed Siboney from Cuba (Sib), as detailed in Table 1 and illustrated in Figure 1. Genomic DNA was extracted using either the Wizard^®^ Genomic DNA Purification Kit (Promega, Madison, WI, USA) or the DNeasy Blood and Tissue Kit (QIAGEN, Hilden, Germany), following the manufacturers’ protocols.

### 2.2. Detection of BLV Proviral Load Using BLV-CoCoMo-qPCR-2

BLV PVLs were determined using a BLV-CoCoMo-qPCR-2 assay (Nippon Gene Co., Ltd., Toyama, Japan), developed based on the “Coordination of Common Motifs” (CoCoMo) algorithm [18,30] using a THUNDERBIRD Probe qPCR Mix (Toyobo, Tokyo, Japan). Briefly, a 183 bp region within the long terminal repeat (LTR) of the BLV genome was amplified using degenerate primers and a FAM-labeled minor groove binder (MGB) TaqMan probe in conjunction with the THUNDERBIRD qPCR Mix (Toyobo, Tokyo, Japan) [35,46]. Simultaneously, a 151 bp fragment of the single-copy bovine leukocyte antigen (BoLA)-*DRA* gene was amplified using a specific primer pair and a FAM-labeled MGB probe, serving as a normalization control for viral genomic DNA quantification [35,46]. Appropriate positive and negative controls were included throughout the assay. The number of proviral copies per 100,000 cells was calculated following the formula established by Jimba et al. [35].

### 2.3. Statistical Analysis

BLV prevalence among the tested cattle was assessed through the direct enumeration of positive and negative cases. Absolute frequencies and categories of BLV proviral load were statistically compared across breeds and countries using Fisher’s exact test based on 10,000 replicates, as implemented in the R statistical environment (https://www.r-project.org/, accessed on 1 August 2020).

### 2.4. Ethical Approval

All animal procedures were reviewed and approved by the Institutional Committee on Care and Use of Experimental Animals (CICUAL) from the School of Veterinary Sciences of the National University of La Plata (Buenos Aires, Argentina; protocols 89-1-18T, 41.2.14T).

## 3. Results

### 3.1. BLV Prevalence of Among Latin American Cattle Breeds

A total of 244 blood samples representing six Latin American cattle breeds were screened for BLV infection using the BLV-CoCoMo-qPCR-2 assay. All DNA samples successfully amplified the *BoLA-DRA* gene, which served as an internal control to verify host DNA integrity and to standardize PVL quantification across samples. BLV provirus was detected in 76 samples, corresponding to an overall prevalence of 31.1% (Table 2 and Table 3), with a mean PVL of 9923 copies per 10^5^ cells (range: 1 to 79,740 copies).

The proportion of BLV-positive animals varied significantly across breeds, ranging from 9.8% in CrAr to 83.8% in CrPaCh (p_among breeds = 9.999 × 10^−5^; Table 2). Fisher’s exact test revealed statistically significant differences in BLV prevalence between CrAr and CrArPat (*p* = 0.003), CrSaa (*p* = 0.001), and CrPaCh (*p* = 1.14 × 10^−8^), as well as between CrPaCh and Siboney (*p* = 0.006).

At the country level, BLV prevalence ranged from 15.6% in Argentina to 83.8% in Paraguay (p_among countries = 9.999 × 10^−5^; Table 3). Significant differences were observed between Argentina and Bolivia (*p* = 0.035), Argentina and Paraguay (*p* = 6.04 × 10^−7^), and between Paraguay and Cuba (*p* = 0.006).

### 3.2. BLV PVL Among Latin American Native Cattle Breeds

PVL is a critical factor influencing both the progression of BLV-associated disease [35,36,37] and the virus’s transmission potential [35,38,39,40,41]. First, PVL was quantified using the BLV-CoCoMo-qPCR-2 assay and summarized PVL in all BLV positive cattle for each sampled breed and country in Latin America (Table 4 and Table 5). Among BLV-positive individuals, a substantial proportion exhibited low PVL levels: 57.9% (44/76) had fewer than 1000 proviral copies per 10^5^ cells, and 13.2% (10/76) fell within the moderate range of 1001 to 9999 copies per 10^5^ cells (Table 4 and Figure 2). Only 28.9% (22/76) of infected animals presented with high PVL (>10,000 copies per 10^5^ cells), predominantly observed in CrPaCh and Sib breeds.

As shown Table 4, comparison of PVL in BLV-positive cattle for each sampled breeds in Latin America showed that Sib breed had the highest percentage of cattle showing high PVL of 10,000 proviral copies per 10^5^ cell and it showed the highest PVL among the six breeds. By contrast, all of three cows in CrCoch breed belonged to PVL levels below 1000 proviral copies per 10^5^ cells and it showed the lowest PVL among the six breeds. As shown in Figure 2, the order of intensity of BLV PVL among breeds was as follows: Sib breed > CrArPat breed > CrAr breed > CrPaCh breed > CrSaa > CrCoch breed.

Next, we summarized PVL in BLV-positive cattle for each sampled county in four Latin America countries (Table 5). Among BLV-positive individuals, three countries, Argentina, Bolivia, and Paraguay, had the highest proportion of cattle group with PVL levels below 1000 proviral copies per 10^5^ cells among low, moderate and high PVL groups. By contrast, Cuba had the highest percentage of cattle group showing high PVL of 10,000 proviral copies per 10^5^ cells. As shown in Figure 3, the order of intensity of BLV PVL among countries was as follows: Cuba > Argentina < Paraguay < Bolivia.

## 4. Discussion

Over the past several decades, BLV prevalence has been investigated across diverse cattle breeds and geographic regions. Polat et al. [2] provided a comprehensive summary of the data published to date. The risk of BLV infection and the associated proviral load are influenced by multiple factors, including geographic location, breed susceptibility (with Taurine breeds generally more vulnerable than Zebu), farm-level variation, production system (dairy herds typically exhibit higher infection rates than beef cattle), animal age, and the year of sampling [47,48,49]. Moreover, the reported prevalence is significantly affected by the diagnostic method employed [47]. Consequently, direct comparisons across studies are challenging and must account for these confounding variables to ensure accurate interpretation.

In the present study, six Latin American cattle breeds were screened for BLV infection using a quantitative PCR assay, revealing an overall prevalence of 31.1%. The proportion of BLV-positive animals varied markedly across breeds, ranging from 9.8% to 83.8%. Consistent with previous reports, high BLV prevalence levels have been documented throughout South America, with enzootic bovine leukosis present in most countries [2,23]. Individual infection rates reported across Latin America span a wide range (from below 20% to over 80%), depending on region, breed, and production system [16,18,19,21,23,24,25,26,50,51,52,53,54,55,56,57,58,59,60,61,62]. Most of these studies have focused on transnational dairy and beef breeds such as Holstein, Angus, and Nellore. In contrast, data on BLV prevalence in native Latin American cattle remain limited. Reports have documented BLV infection in various Creole cattle populations and local composite breeds from Bolivia, Colombia, Brazil, and Panama [21,23,26,44,59,60,61,62]. These findings are summarized in Table 6. However, only one of these studies employed quantitative methods [23], with the majority relying on qualitative detection, thereby limiting insights into viral load and disease progression.

A comparative analysis of BLV prevalence across native cattle breeds and countries reveals substantial variability, ranging from 0% to over 90% (Table 6). The elevated infection rates observed in CrPaCh, CrSaa and CrArPat are consistent with the widespread circulation of BLV in Paraguay, Bolivia and Argentina, particularly among dairy breeds such as Holstein [23,38]. Polat et al. [23] reported average Holstein farm-level prevalences of 77.4% in Argentina, 65.3% in Bolivia, and 54.7% in Paraguay, respectively. Comparable prevalence levels have been reported in other Creole breeds, including Hartón del Valle (83.3%) and Chino Santadereano (60%) from Colombia and Guaymí (8%) from Panama [21,61]. Moderate infection rates were recorded in CrCoch (23.8%) and Yacumeño (20.37%) from Bolivia. These figures are notably lower than those observed in the Bolivian dual-purpose CrSaa breed (70%), which may reflect the influence of more intensive dairy production practices. However, this hypothesis warrants validation through studies with larger sample sizes. The CrAr population from Argentina exhibited a low BLV infection rate, estimated at 9.8%. Additional data from Creole breeds in Brazil, Colombia, and Ecuador indicate intermediate prevalence levels ranging from 10% to 35% (Table 6). The Siboney breed from Cuba, a dairy composite developed in the 1970s through the crossbreeding of Holstein cattle with local Zebu to enhance milk production under tropical conditions, exhibited a relatively lower BLV prevalence (28.6%) compared to other composite breeds from Colombia, such as Lucerna and Velásquez, both with reported prevalences of 50% [21,44]. This reduced infection rate in Siboney may be partially attributable to the genetic contribution of Zebu (*Bos indicus*), which has been suggested to confer greater resistance to BLV, potentially mitigating the susceptibility associated with Taurine (*Bos taurus*) ancestry. However, the Velásquez breed, which includes approximately 25% Brahman genetics, also showed a prevalence of 50%, indicating that Zebu ancestry alone may not fully account for resistance. Additionally, Creole breeds across Latin America exhibit variable levels of *Bos indicus* introgression, typically lower in temperate and cold environments and higher in subtropical and tropical regions [63,64,65]. Nonetheless, current BLV prevalence data in Creole breeds from different countries do not reveal a consistent correlation between infection levels and the proportion of Zebu ancestry.

The BLV-CoCoMo-qPCR assay offers distinct advantages over conventional diagnostic methods. First, it enables the detection of a broad spectrum of BLV strains, including both known and potentially novel variants [35]. Second, it has enhanced sensitivity compared to that of other methods because it detects the BLV LTR region, which is present at two copies per provirus [35,66,67]. In previous study, our group evaluated the BLV provirus detection limit using BLV-infectious molecular clones (pBLV-IF2) by comparing to other two BLV proviral quantitative real-time PCR methods, such as TaqMan MGB assay targeting pol gene [68,69], and Cycleave assay targeting the tax gene [37]. The result shows that the BLV-CoCoMo-qPCR assay could detect 100% (3/3) of pBLV-IF2 when present at 0.78125 copies per 10^5^ cells, indicating that the BLV-CoCoMo-qPCR assay has a high sensitivity [67]. Thus, analytical sensitivity using BLV-infectious molecular clones of the BLV-CoCoMo-qPCR assay was found to be highly sensitive when compared with other real-time PCRs. Furthermore, we have successfully detected provirus in low-copy cows at one copy per 10^5^ cells using 370 field cattle [70]. Currently, we show that the BLV-CoCoMo-qPCR assay consistently detected BLV proviruses in low-copy cows at around 10 copies per 10^5^ cells using 82 field cattle [71]. Thus, the BLV-CoCoMo-qPCR assay has a high sensitivity for BLV provirus detection in diagnostic analysis using field samples. This discrepancy is largely attributable to sequence mismatches in the primer annealing regions, a common source of false negatives in conventional PCR and serological assays. The use of degenerate primers in the CoCoMo-qPCR assay effectively mitigates this issue. Importantly, BLV-CoCoMo-qPCR is a quantitative method that allows for precise estimation of proviral load, a critical parameter often overlooked in earlier prevalence studies, which typically reported only the proportion of infected animals. PVL has been positively correlated with the clinical progression of EBL, with higher BLV copy numbers associated with increased disease severity [35]. Moreover, PVL quantification may aid in identifying animals with natural resistance to BLV, offering a valuable tool for genetic studies aimed at uncovering markers of susceptibility or resilience.

As previously noted, BLV was detected across all studied breeds, with prevalence rates ranging from 9.8% to 83.8%. Notably, a substantial proportion of BLV-positive animals exhibited low proviral loads (PVL), with 57.9% harboring fewer than 1000 copies per 10^5^ cells and 13.2% between 1001 and 9999 copies per 10^5^ cells (Table 4 and Table 5). The highest proportions of animals with PVL exceeding 1000 copies were observed in the CrArPat and Siboney breeds. These findings suggest the potential presence of BLV-resistant individuals within certain Creole populations. Supporting this hypothesis, Hernández-Herrera et al. [44] reported that Hartón del Valle Creole cattle not only exhibited lower BLV infection rates but also showed reduced lymphocytosis, a more robust immune response, and significantly lower PVL compared to the Lucerna composite breed and Holstein. Collectively, these observations reinforce the notion that genetic resistance to elevated PVL and progression to enzootic bovine leukosis (EBL) may exist in specific Creole breeds. While multiple studies in transnational breeds such as Holstein have consistently demonstrated a strong association between low PVL and reduced risk of disease progression, further clinical and longitudinal data in Creole cattle are needed to validate this resistance hypothesis in Latin American native breeds. Such evidence could support the inclusion of PVL-based resistance traits in selective breeding programs aimed at improving herd resilience and mitigating BLV-associated economic losses.

## 5. Conclusions 

This study provides the first comprehensive assessment of BLV prevalence and PVL in native Latin American native cattle populations across a broad geographical range, using the BLV-CoCoMo-qPCR-2 assay. These findings offer novel insights into BLV epidemiology in underrepresented genetic backgrounds and lay the groundwork for future surveillance and control strategies in the region.

## Figures and Tables

**Figure 1 pathogens-14-01005-f001:**
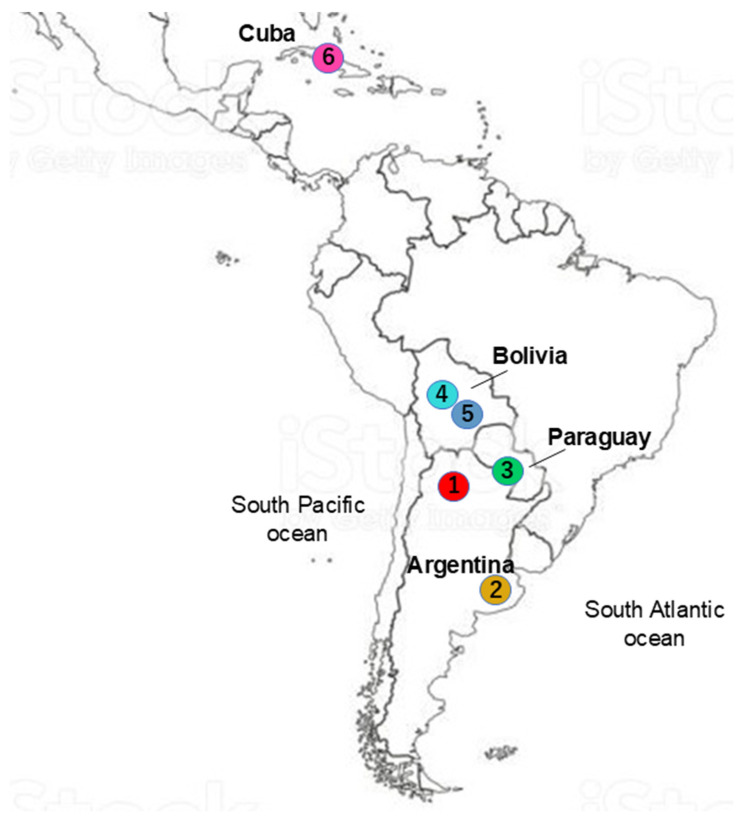
Sampling sites: 1. Argentine Creole (CrAr), 2. Patagonian Argentine Creole (CrArPat), 3. Pampa Chaqueño Creole from Paraguay (CrPaCh), 4. Bolivian Creole from Cochabamba department (CrCoch), 5. Saavedreño Creole (CrSaa), and 6. the Siboney from Cuba composite breed (Sib).

**Figure 2 pathogens-14-01005-f002:**
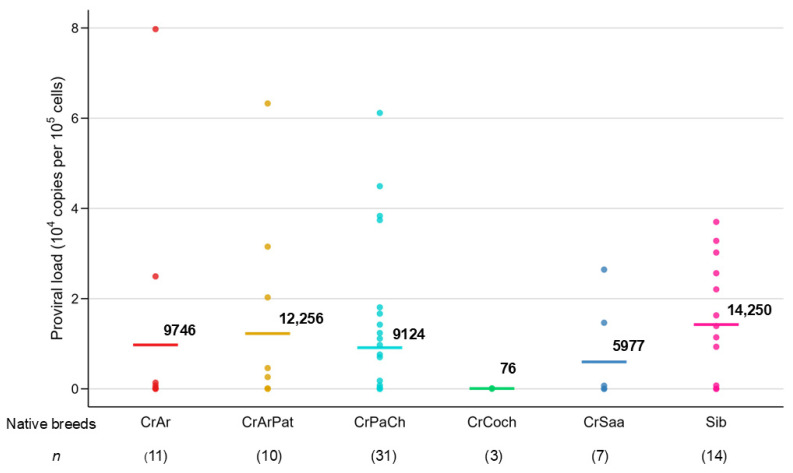
BLV infectious in six Latin American native cattle populations using the CoCoMo2 qPCR method. Argentine Creole (CrAr), Patagonian Argentine Creole (CrArPat), Pampa Chaqueño Creole from Paraguay (CrPaCh), Bolivian Creole from Cochabamba department (CrCoch), Saavedreño Creole (CrSaa), and the Siboney from Cuba composite breed (Sib).

**Figure 3 pathogens-14-01005-f003:**
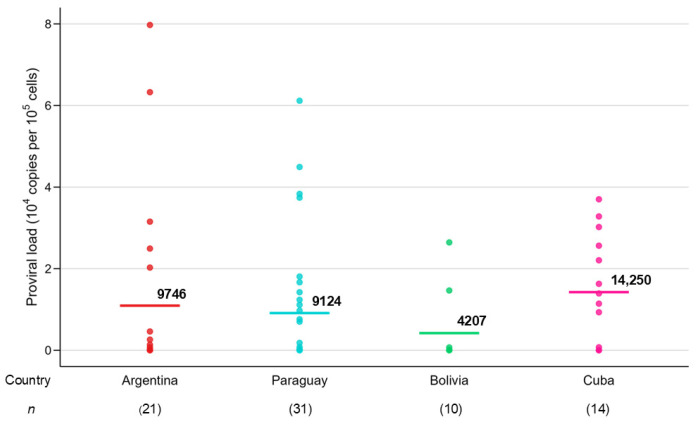
BLV infectious in four Latin American countries using the CoCoMo2 qPCR method. Argentina, Paraguay, Bolivia and Cuba.

**Table 1 pathogens-14-01005-t001:** Main characteristic of the studied native breed/population.

Breed	Sample Size	Breed Origin	Sample Site	Production Purpose	Environment
Argentine Creole (CrAr)	112	Argentina	Tucumán province	Beef	Subtropical to temperate
Argentine Patagonian Creole	23	Argentina	Buenos Aires province	Beef	Temperate to cold
(CrArPat)					
Bolivian Highland Creole (CrCoch)	13	Bolivia	Cochabamba department	Beef	Highland plain
Saavedreño Creole (CrSaa)	10	Bolivia	Santa Cruz department	Beef and dairy	Subtropical plain
Pampa Chaqueño Creole (CrPaCh)	37	Paraguay	Paraguayan Chaco	Beef	Dry forest
Siboney de Cuba (Sib)	49	Cuba	La Havana province, Cuba	Dairy	Subtropical plain

**Table 2 pathogens-14-01005-t002:** Detection of BLV infection in six native bovine breeds from Argentina, Bolivia, Paraguay, and Cuba by CoCoMo-quantitative PCR (qPCR).

Breed	Positive No. ^a^/Tested Samples No. (Positive %)
Argentine Creole (CrAr)	11/112 (9.8)
Argentine Patagonian Creole (CrArPat)	10/23 (43.5)
Saavedreño Creole (CrSaa)	7/10 (70.0)
Bolivian Highland Creole (CrCoch)	3/13 (23.8)
Pampa Chaqueño (CrPaCh)	31/37 (83.8)
Siboney (Sib)	14/49 (28.6)
Total breeds	76/244 (31.1)

^a^ no = number; CrAr = Argentine Creole; CrArPat = Argentine Patagonian Creole; CrCoch = Cochabamba; Saavedreño (CrSaa); Pampa Chaqueño (CrPaCh); Siboney (Sib).

**Table 3 pathogens-14-01005-t003:** Detection of BLV infection in native cattle from Argentina, Bolivia, Paraguay, and Cuba by CoCoMo-quantitative PCR (qPCR).

Country	Positive No.^a^/Tested Samples No. (Positive %)
Argentina	21/135 (15.6)
Bolivia	10/23 (43.5)
Paraguay	31/37 (83.8)
Cuba	14/49 (28.6)
Total breeds	76/244 (31.1)

^a^ no = number.

**Table 4 pathogens-14-01005-t004:** BLV prevalence and proviral load estimated using the BLV-CoCoMo-qPCR-2 assay in six local cattle breeds/populations from Latin American, including Argentine Creole (CrAr), Patagonian Argentine Creole (CrArPat), Pampa Chaqueño Creole from Paraguay (CrPaCh), Bolivian Creole from Cochabamba department (CrCoch), Saavedreño Creole (CrSaa), and the Siboney from Cuba composite breed (Sib).

Breed	BLV– Positive *n* ^a^	PVL ^b^ (Copy/10^5^ Cells)			Mean Value (Range)
		1–1000 *n* (%)	1001–9999 *n* (%)	≥10,000 *n* (%)	
CrAr	11	8 (72.7)	1 (9.1)	2 (18.2)	9746 (3–79,740)
CrArPat	10	5 (50.0)	2 (20.0)	3 (30.0)	12,256 (13–63,263)
CrSaa	7	5 (71.4)	2 (28.6)	0 (0.0)	5977 (5–26,435)
CrCoch	3	3 (100.0)	0 (0.0)	0 (0.0)	76 (21–119)
CrPaCh	31	18 (58.1)	4 (12.9)	9 (29.0)	9124 (1–61,161)
Sib	14	5 (35.7)	1 (7.2)	8 (57.1)	14,250 (1–37,009)
Total	76	44 (57.9)	10 (13.2)	22 (28.9)	9923 (1–79,740)

^a^ *n* = number of cattle; ^b^ PVL = proviral load.

**Table 5 pathogens-14-01005-t005:** BLV prevalence and proviral load estimated using the BLV-CoCoMo-qPCR-2 assay in local cattle breeds/populations from Argentina, Bolivia, Paraguay, and Cuba.

Country	BLV– Positive *n* ^a^	PVL ^b^ (Copy/10^5^ Cells)			Mean Value (Range)
		1–1000 *n* (%)	1001–9999 *n* (%)	≥10,000 *n* (%)	
Argentina	21	13 (61.9)	3 (14.29)	5 (23.8)	9746 (3–79,740)
Bolivia	10	8 (80.0)	2 (20.0)	0 (0.0)	4207 (5–26,435)
Paraguay	31	18 (58.1)	4 (12.9)	9 (29.0)	9124 (1–61,161)
Cuba	14	5 (35.7)	1 (7.2)	8 (57.1)	14,250 (1–37,009)
Total	76	44 (57.9)	10 (13.2)	22 (29.0)	9923 (1–79,740)

^a^ *n* = number of cattle; ^b^ PVL = proviral load.

**Table 6 pathogens-14-01005-t006:** Comparison of BLV prevalence among Creole cattle and local composite breeds from Latin America.

Breed	Typed	Country	Used Method	*n* ^a^	BLV ^b^ Prevalence (%)	Reference
Argentine Creole	Creole	Argentina	CoCoMo qPCR	112	9.8 (11/112)	This work
Argentine Patagonian Creole	Creole	Argentina	CoCoMo qPCR	23	43.5 (10/23)	This work
Pampa Chaqueño	Creole	Paraguay	CoCoMo qPCR	37	83.2 (31/37)	This work
Saavedreño	Creole	Bolivia	CoCoMo qPCR	10	70 (7/10)	This work
Cochabamba Creole	Creole	Bolivia	CoCoMo qPCR	13	23.1 (3/13)	This work
Yacumeño	Creole	Bolivia	CoCoMo qPCR	130	20.37 (22/108)	[23]
Lageano	Creole	Brazil	standard PCR	308	36.69 (113/308)	[60]
Curraleiro Pé-Duro	Creole	Brazil	ELISA	596	18.29 (109/596)	[59]
Blanco Orijinegro	Creole	Colombia	nested PCR	30	0 (0/30)	[21]
Casareño	Creole	Colombia	nested PCR	30	26.7 (8/30)	[21]
Casareño	Creole	Colombia	nested PCR	73	16.4 (12/73)	[26]
Costeño con Cuernos	Creole	Colombia	nested PCR	30	23.3 (7/30)	[21]
Chino Santandereano	Creole	Colombia	nested PCR	30	60 (18/30)	[21]
Chino Santandereano	Creole	Colombia	nested PCR	108	35.1 (38/108)	[26]
Caqueño	Creole	Colombia	nested PCR	30	16.7 (5/30)	[21]
Hartón del Valle	Creole	Colombia	nested PCR	30	83.3 (25/30)	[21]
Hartón del Valle	Creole	Colombia	nested PCR, qPCR	93	18.9 (18/93)	[44]
Romosinuano	Creole	Colombia	nested PCR	30	0 (0/30)	[21]
Sanmartineano	Creole	Colombia	nested PCR	30	0 (0/30)	[21]
Sanmartineano	Creole	Colombia	nested PCR	72	23.6 (17/72)	[26]
Peruvian Creole	Creole	Perú	AGID	nd	nd	[24]
Siboney	Composite ^c^	Cuba	CoCoMo qPCR	49	28.6 (14/49)	This work
Lucerna	Composite ^d^	Colombia	nested PCR	30	50 (15/30)	[21]
Lucerna	Composite ^d^	Colombia	nested PCR, qPCR	24	48.5 (17/72)	[44]
Velásquez	Composite ^e^	Colombia	nested PCR	30	50 (15/30)	[21]
Creole	Creole	Ecuador	ELISA	1604	10.6 (170/1604)	[62]
Guaymí	Creole	Panamá	nested PCR	40	80 (32/40)	[61]

^a^ *n* = number of cattle, ^b^ PVL = proviral load, ^c^ 5/8 Holstein Friesian—3/8 local Zebu, ^d^ 40% Friesian—30% Hartón—30% Shorthorn, and ^e^ 40% Red Poll—25% Brahman—25% Romosinuano. nd = not determined.

## Data Availability

The datasets generated during and/or analyzed during the current study are available from the corresponding authors on reasonable request.

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
