# Peer review of "Detection of Bovine Leukemia Virus in Argentine, Bolivian, Paraguayan and Cuban Native Cattle Using a Quantitative Real-Time PCR Assay-BLV-CoCoMo-qPCR-2"

_pathogens, 2025, doi:10.3390/pathogens14101005_

Round 1
Reviewer 1 Report
Comments and Suggestions for Authors
The manuscript is generally very well described. The background of the research team contributes greatly to the development of this work. International cooperation is noteworthy.
Regarding the format, Figures 2 and 3 need to be reviewed, as they appear to be duplicated. I do not fully agree with the title, as it suggests broader coverage of Latin American countries; I suggest specifying "6 Latin American countries."
On the other hand, the discussion devotes many lines to the technique employed, and while this is important, I do not consider it the main focus of the study.
I´m concerned about the use of the term "prevalence" in the title and results, since in all cases there is no description of the total inventory of these animals in each country, so calling it "prevalence" seems somewhat ambitious.
Finally, this type of study requires approval from an ethics committee, which is not mentioned anywhere in the manuscript.
Author Response
Thank you very much for taking the time to review this manuscript. Please find the detailed responses below and the corresponding revisions/corrections highlighted/in track changes in the re-submitted files.
Reviewer 1
Comments and Suggestions for Authors
The manuscript is generally very well described. The background of the research team contributes greatly to the development of this work. International cooperation is noteworthy.
Comments 1: Regarding the format, Figures 2 and 3 need to be reviewed, as they appear to be duplicated. I do not fully agree with the title, as it suggests broader coverage of Latin American countries; I suggest specifying "6 Latin American countries."
Response 1: The title was modified according to the Reviewer 1 suggestion. Sorry for the mistake, the correct figure 2 was included in the new version.
Comments 2: On the other hand, the discussion devotes many lines to the technique employed, and while this is important, I do not consider it the main focus of the study.
Response 2: Yes, we agree with Reviewer 1 that the discussion section contains several statements highlighting the advantages of the technique employed in comparison to those used in previous studies. However, we consider this emphasis to be essential for contextualizing and interpreting the present findings relative to previously reported data. The methodological differences directly influence detection sensitivity and accuracy, and therefore merit detailed discussion, and limiting the comparison with previous reports.
Comments 3: I´m concerned about the use of the term "prevalence" in the title and results, since in all cases there is no description of the total inventory of these animals in each country, so calling it "prevalence" seems somewhat ambitious.
Response 3: The term “Prevalence” was replaced by “Detection” in the title following the Reviewer 1 suggestion.
Comments 4: Finally, this type of study requires approval from an ethics committee, which is not mentioned anywhere in the manuscript.
Response 4: The required information was added in the new version: section “2.4. Ethical Approval”.
Reviewer 2 Report
Comments and Suggestions for Authors
The paper entitled “Prevalence of bovine leukemia virus in Latin American native 2 cattle using a quantitative real-time PCR assay-BLV-CoCoMo-3 qPCR-2” described the higher prevalence in different breeds and countries. The contribution is important to increase the epidemiology of this virus in Latin America and worldwide. Although, the authors need to be more careful with the description and review of the results, presenting many errors in the writing and presentation of tables and figures. Additionally, is necessary to improve the discussion of the manuscript.
Introduction
Line 59 – You write that the first description was in 1878, although the reference is from 1871. What is the correct year?
Line 83 - Boophilus microplus must be in italic
Line 103-105: This sentence was result and conclusion, not objective. I suggest to remove from this section
Materials and Methods
Line 124 - in this subsection (2.2) was not clear the references of the BLV amplification.
- 183 bp region within the long terminal repeat (LTR) (reference?)
- 151 bp fragment of the single-copy bovine leukocyte antigen 131 (BoLA)-DRA gene (reference?)
Line 136 – The sentence “conclusions that can be drawn.” does not make sense in this paragraph
Results
Line 157 – Remove the ( . ) in “(Tables 2 and 3).,”
Line 160 – “prevalence rates spanning from 15.8%”, in the table is describe 15.6%. Please correct the percentage.
Line 160 – The statistical analysis described only here, does not described the difference between which breed. Is necessary to demonstrated in the table. Who is different from who?
And between countries, did you do the analyses?
Line 172 – Please, rewrite the sentence.
Line 178 – Again, the percentage in the text is different that described in the table 4. Please determine what is the correct.
Line 179 – the prevalence described is the absolute number or the percentage? Because the CrArPat presents 30% of animals in the higher PVL
To limited the uncomprehensive situation, I strongly recommend to use the absolute number/total number after the percentages in the text (example in line 177: “and 13.2% (10/76) fell within …”)
Figure 2 – the numbers (viral load and positives) in this figure were distributed different from the numbers in the table 4 (see CrCoch and CrPaCh). I believe the table 5 is the correct one.
Line 202 – the order of intensity of BLV PVL among breeds is incorrect. These two breeds (CrArPat/CrSaa) were not similar with each other. Please correct the sequence “Sib breed > CrArPat/CrSaa breeds > CrAr breed > CrPaCh breed > CrCoch breed.”
Line 207 – remove the (.)
Line 209 - 105 cells?
Line 209 – 211 – The sentence describes the table exactly. I suggest to removing it.
Figure 3 is the same Figure 2. Be careful.
Discussion
The discussion is really confused. You have to review the information and the description on all the topic.
Line 220 – Do you believe that a reference from 2017 provides a comprehensive summary of the data published to date? There hasn't been anything more recent than that reference?
Line 230 – The overall positive is 76 / 244 (31.1). Not 32.4%
Line 232 – “relatively high BLV prevalence levels” - What is the value of this high prevalence? And in the end of the sentence there were only one author, there were other studies like yours in South America?
Line 229-242 – This paragraph could be explored better, there is a lack of comparison between the results in Creoles and your findings. The information was superficial.
Table 6 – Some errors were detected
- In “Typed” – there are letters in Composite, but they are not in the legend, and the letters in the legend were wrong if compared with the table.
- In “Country” - Cuba need capital letter.
- In “BLV prevalence (%)” – I recommended including the absolute number/total number of samples from all studies, to say how representative these percentages are.
- According to the table, there were only nine studies with Creole breeds. You can explore these researches.
Line 248 – Need space prior the sentence
Line 249 – Where is the 90%? The table is 3 or 6?
Line 249-251 – CrPaCh have 83.8% and CrArPat 43.5%, What is the prevalence described in Holstein breed to compare?
What you consider high, moderate and low prevalence. The text was really confused. I disagree that 9.8% from Argentine is moderate prevalence (misspelled as ArCr in line 254).
Line 260 – You considered 28.6% as lower prevalence, and 9.8% was considered moderate before. Why the comparation with Colombia if this information is about Cuba?
Line 268 – 275 – These sentences were confused, need to be rewrite
Reference 47 and 48 – You must refer these references as our group, not we.
Lines 283 e 288 – In these two lines you compare the molecular detection with serological detection. In these techniques different targets were detected, making impossible the comparison.
Paragraph 264-296 – Is to long, and the citations were all from your research group. You need to improve the discussion about all your results, inclusive with the other reference presented in table 6.
Comments on the Quality of English Language
Please send the manuscript for professional proofreading.
Author Response
Thank you very much for taking the time to review this manuscript. Please find the detailed responses below and the corresponding revisions/corrections highlighted/in track changes in the re-submitted files.
Reviewer 2
Comments and Suggestions for Authors
The paper entitled “Prevalence of bovine leukemia virus in Latin American native 2 cattle using a quantitative real-time PCR assay-BLV-CoCoMo-3 qPCR-2” described the higher prevalence in different breeds and countries. The contribution is important to increase the epidemiology of this virus in Latin America and worldwide. Although, the authors need to be more careful with the description and review of the results, presenting many errors in the writing and presentation of tables and figures. Additionally, is necessary to improve the discussion of the manuscript.
Introduction
Comments 1: Line 59 – You write that the first description was in 1878, although the reference is from 1871. What is the correct year?
Response 1: The correct year is 1871. This was modified in the new version.
Comments 2: Line 83 - Boophilus microplus must be in italic.
Response 2: This was done in the new version.
Comments 3: Line 103-105: This sentence was result and conclusion, not objective. I suggest to remove from this section
Response 3: This sentence was deleted following the Revier 2 suggestion.
Materials and Methods
Comments 4: Line 124 - in this subsection (2.2) was not clear the references of the BLV amplification.
- 183 bp region within the long terminal repeat (LTR) (reference?)
- 151 bp fragment of the single-copy bovine leukocyte antigen 131 (BoLA)-DRA gene (reference?)
Response 4: This information is described in detail in reference 18 and 30 that were cited in this section.
Comments 5: Line 136 – The sentence “conclusions that can be drawn.” does not make sense in this paragraph
Response 5: This sentence was deleted.
Results
Comments 6: Line 157 – Remove the ( . ) in “(Tables 2 and 3).,”
Response 6: This was done.
Comments 7: Line 160 – “prevalence rates spanning from 15.8%”, in the table is describe 15.6%. Please correct the percentage.
Response 7: This typing error was corrected.
Comments 8: Line 160 – The statistical analysis described only here, does not described the difference between which breed. Is necessary to demonstrated in the table. Who is different from who?
And between countries, did you do the analyses?
Response 8: The statistical analysis was performed among and between pair of breeds and countries. This was detailed and clarified in the new version.
Comments 9: Line 172 – Please, rewrite the sentence.
Response 9: This sentence was rewritten.
Comments 10: Line 178 – Again, the percentage in the text is different that described in the table 4. Please determine what is the correct.
Response 10: This typing error was corrected.
Comments 11: Line 179 – the prevalence described is the absolute number or the percentage? Because the CrArPat presents 30% of animals in the higher PVL
To limited the uncomprehensive situation, I strongly recommend to use the absolute number/total number after the percentages in the text (example in line 177: “and 13.2% (10/76) fell within …”)
Response 11: In the text the prevalence described was percentage. In the new version the suggested changes were introduced.
Comments 12: Figure 2 – the numbers (viral load and positives) in this figure were distributed different from the numbers in the table 4 (see CrCoch and CrPaCh). I believe the table 5 is the correct one.
Response 12: Figure 2 was corrected.
Comments 13: Line 202 – the order of intensity of BLV PVL among breeds is incorrect. These two breeds (CrArPat/CrSaa) were not similar with each other. Please correct the sequence “Sib breed > CrArPat/CrSaa breeds > CrAr breed > CrPaCh breed > CrCoch breed.”
Response 13: This was corrected in the new version.
Comments 14: Line 207 – remove the (.)
Response 14: The “.” Was removed.
Comments 15: Line 209 - 105 cells?
Response 15: “105” was replace by “105”
Comments 16: Line 209 – 211 – The sentence describes the table exactly. I suggest to removing it.
Response 16: This sentence was deleted following the Reviewer 2 suggestion.
Comments 17: Figure 3 is the same Figure 2. Be careful.
Response 17: Yes, you are right. Sorry for the mistake. The figure 3 was replace for the correct one.
Discussion
The discussion is really confused. You have to review the information and the description on all the topic.
Comments 18: Line 220 – Do you believe that a reference from 2017 provides a comprehensive summary of the data published to date? There hasn't been anything more recent than that reference?
Response 18: To the best of our knowledge, this article presents the most comprehensive epidemiological summary of BLV data published to date. Following its publication, several additional studies have focused on specific cattle breeds within individual countries or regions. Moreover, phylogenetic analyses of BLV strains have since been reported,
Comments 19: Line 230 – The overall positive is 76 / 244 (31.1). Not 32.4%
Response 19: This was corrected.
Comments 20: Line 232 – “relatively high BLV prevalence levels” - What is the value of this high prevalence? And in the end of the sentence there were only one author, there were other studies like yours in South America?
Response 20: The term relatively was deleted to avoid confusion.
Comments 21: Line 229-242 – This paragraph could be explored better, there is a lack of comparison between the results in Creoles and your findings. The information was superficial.
Response 21: In the new version, more details about the comparison between our results and previous reports were added.
Comments 22: Table 6 – Some errors were detected
- In “Typed” – there are letters in Composite, but they are not in the legend, and the letters in the legend were wrong if compared with the table.
- In “Country” - Cuba need capital letter.
- In “BLV prevalence (%)” – I recommended including the absolute number/total number of samples from all studies, to say how representative these percentages are.
- According to the table, there were only nine studies with Creole breeds. You can explore these researches.
Response 22: The super index letters in Composite breeds were explained in the footnote of the table. The capital letter for Cuba was corrected. The absolute number/total values were added in the”BLV prevalence (=%) column.
Comments 23: Line 248 – Need space prior the sentence.
Response 23: This space was added.
Comments 24: Line 249 – Where is the 90%? The table is 3 or 6?
Response 24: This value referred to Table 6. This was corrected.
Comments 25: Line 249-251 – CrPaCh have 83.8% and CrArPat 43.5%, What is the prevalence described in Holstein breed to compare?
What you consider high, moderate and low prevalence. The text was really confused. I disagree that 9.8% from Argentine is moderate prevalence (misspelled as ArCr in line 254).
Response 25: The prevalence in Holstein correspond to reference [12] that it is cited in this sentence. This work analysed BLV prevalence in Holstein populations from Argentine (77.4%, 325/420), Bolivia (56.3%, 90/160), Chile (27.8, 98/352), Paraguay (54.7%), and Perú (45.9%, 61/133) (See Table 1 of this work). This sentence was rephrased in order to clarified. ArCr in line 254 was corrected.
Comments 26: Line 260 – You considered 28.6% as lower prevalence, and 9.8% was considered moderate before. Why the comparation with Colombia if this information is about Cuba?
Response 26: The sentence began “Moderate to low infection rates were recorded in..” However, this sentence was rewritten in order to clarify.
Comments 27: Line 268 – 275 – These sentences were confused, need to be rewrite
Response 27: This sentence was rewritten.
Comments 28: Reference 47 and 48 – You must refer these references as our group, not we.
Response 28: term “we” was replaced by “our group”.
Comments 29: Lines 283 e 288 – In these two lines you compare the molecular detection with serological detection. In these techniques different targets were detected, making impossible the comparison.
Response 29: Comparison with serological technique was deleted.
Comments 30: Paragraph 264-296 – Is to long, and the citations were all from your research group. You need to improve the discussion about all your results, inclusive with the other reference presented in table 6.
Response 30: The four other references have added in this discussion section.
Comments on the Quality of English Language
Please send the manuscript for professional proofreading.
Reviewer 3 Report
Comments and Suggestions for Authors
This manuscript reports the prevalence of bovine leukemia virus (BLV) and proviral load (PVL) in six native or composite cattle breeds across Argentina, Bolivia, Paraguay, and Cuba. Using the BLV-CoCoMo-qPCR-2 assay, the authors examined 244 animals and found an overall prevalence of 31.1%. Infection rates varied significantly by breed and country, ranging from 9.8% to 83.8%. Notably, more than half of the BLV-positive animals exhibited low PVL, suggesting a potential resistance to enzootic bovine leukosis (EBL) progression. The study represents the first comprehensive analysis of BLV PVL in Creole cattle across Latin America and provides valuable epidemiological insights into underrepresented genetic resources.
Minor Comments
1. Creole breeds are known to have variable levels of admixture with Zebu and other breeds, which could influence BLV susceptibility. The Discussion would be strengthened by addressing how breed history and genetic background might impact the observed prevalence and PVL.
2. The authors suggest that low PVL indicates potential resistance to EBL progression. However, no clinical or longitudinal data are provided to substantiate this. The claim of “resistance” should be toned down or clearly framed as a hypothesis, and the absence of clinical correlation should be acknowledged as a limitation.
3. Both “provirus load” and “proviral load” appear in the text. Please standardize terminology throughout the manuscript.
Author Response
Thank you very much for taking the time to review this manuscript. Please find the detailed responses below and the corresponding revisions/corrections highlighted/in track changes in the re-submitted files.
Reviewer 3
Comments and Suggestions for Authors
This manuscript reports the prevalence of bovine leukemia virus (BLV) and proviral load (PVL) in six native or composite cattle breeds across Argentina, Bolivia, Paraguay, and Cuba. Using the BLV-CoCoMo-qPCR-2 assay, the authors examined 244 animals and found an overall prevalence of 31.1%. Infection rates varied significantly by breed and country, ranging from 9.8% to 83.8%. Notably, more than half of the BLV-positive animals exhibited low PVL, suggesting a potential resistance to enzootic bovine leukosis (EBL) progression. The study represents the first comprehensive analysis of BLV PVL in Creole cattle across Latin America and provides valuable epidemiological insights into underrepresented genetic resources.
Minor Comments
Comments 1: Creole breeds are known to have variable levels of admixture with Zebu and other breeds, which could influence BLV susceptibility. The Discussion would be strengthened by addressing how breed history and genetic background might impact the observed prevalence and PVL.
Response 1: We appreciate Reviewer 3’s observation regarding the potential interbreeding between Creole cattle populations and Zebu. This topic has been previously investigated by our research group, which demonstrated that native populations from Argentina, Bolivia, and Paraguay exhibit low levels of Bos indicus introgression. These estimates were derived using both autosomal markers (STRs and SNPs) and uniparental markers (e.g., Liron et al., 2006; Ginja et al., 2019; Marcuzzi et al., 2025). In contrast, the Siboney breed from Cuba is a composite population with a defined genetic structure: 5/8 Holstein Friesian and 3/8 local Zebu ancestry. We have incorporated this important consideration into the revised version of the manuscript.
Comments 2: The authors suggest that low PVL indicates potential resistance to EBL progression. However, no clinical or longitudinal data are provided to substantiate this. The claim of “resistance” should be toned down or clearly framed as a hypothesis, and the absence of clinical correlation should be acknowledged as a limitation.
Response 2: We agree with Reviewer 3 that clinical evidence supporting the statement "low PVL indicates potential resistance to EBL progression" in native Latin American cattle breeds remains limited. In contrast, multiple studies conducted in transnational breeds such as Holstein have consistently demonstrated an association between low PVL and reduced risk of disease progression. In accordance with the reviewer’s suggestion, we have revised the sentence to reflect a more cautious and hypothetical tone, acknowledging the current lack of breed-specific clinical data.
Comments 3: Both “provirus load” and “proviral load” appear in the text. Please standardize terminology throughout the manuscript.
Response 3: The term “provirus load” was replaced by “proviral load” to standardize terminology.
Round 2
Reviewer 2 Report
Comments and Suggestions for Authors
Comments and Suggestions for Authors
The review of the paper entitled “Prevalence of bovine leukemia virus in Latin American native 2 cattle using a quantitative real-time PCR assay-BLV-CoCoMo-3 qPCR-2” improve significantly the quality of the research, making publication possible.
Only two corrections are needed:
Table 5: Change proviral to PVLb, similar to table 4.
Figure 3: The number from Argentina was different from the table 5. In figure 3 was 9746 and in the table 5 was 10941.
Author Response
Comments and Suggestions for Authors
The review of the paper entitled “Prevalence of bovine leukemia virus in Latin American native 2 cattle using a quantitative real-time PCR assay-BLV-CoCoMo-3 qPCR-2” improve significantly the quality of the research, making publication possible.
Only two corrections are needed:
Comments 1: Table 5: Change proviral to PVLb, similar to table 4.
Response 1: This chance was done
Comments 2: Figure 3: The number from Argentina was different from the table 5. In figure 3 was 9746 and in the table 5 was 10941.
Comments 2: The value was corrected in table 5.